# Strengthening regulation for medical products in Tanzania: An assessment of regulatory capacity development, 1978–2020

Adam M. Fimbo[1], Hiiti B. Sillo[2], Alex Nkayamba[1], Sunday Kisoma[1], Yonah Hebron Mwalwisi[1], Rafiu Idris[3], Sarah Asiimwe[3], Patrick Githendu[3], Osondu Ogbuoji[4], Linden Morrison[3], Jesse B. Bump[5], Eliangiringa Kaale[6] *

1 Directorate of Medicine Control, Tanzania Medicines and Medical Devices Authority, Dar es Salaam, Tanzania, 2 Regulation and Safety, Regulation and Prequalification Department, World Health Organization, Geneva, Switzerland, 3 High Impact Africa 2 Department, The Global Fund to Fight AIDS, Tuberculosis and Malaria, Geneva, Switzerland, 4 Duke Global Health Institute, Duke University, Durham, North Carolina, United States of America, 5 Global Health and Population, Harvard T.H. Chan School of Public Health, Boston, Massachusetts, United States of America, 6 Department of Medicinal Chemistry and The Pharm R&D Laboratory, School of Pharmacy, Muhimbili University of Health and Allied Sciences, Dar es Salaam, Tanzania

* eliangiringa.kaale@muhas.ac.tz

**Data Availability Statement:** All relevant data are within the paper.

## Abstract

Improving medicines regulation can lead to better population health, but how this process works in low- and middle-income countries remains underexplored. Tanzania's pharmaceutical sector is often cited as a successful example of a well-functioning regulatory system in a developing country, attributed to the work of the Tanzania Food and Drugs Authority (TFDA), now the Tanzania Medicines and Medical Devices Authority (TMDA). This raises the question: how was this regulatory capacity developed, and what lessons can other countries learn from Tanzania's experience? This paper analyzes changes in Tanzania's pharmaceutical regulation over three periods of significant sectoral reform. A desk review was conducted of Tanzania's policies, laws, regulations, guidelines, procedures, and institutional reports. The study reveals that Tanzania's regulatory capacity improved significantly through targeted reforms that addressed challenges in key regulatory areas. The three key periods examined are: 1) The separation of medicines regulation from food safety (1978–2003), 2) The expansion of regulatory domains and the establishment of a semi-autonomous regulatory agency (2003–2011), and 3) The expanded role of the Pharmacy Council to include premises regulation (2011–2020). The development of a well-functioning regulatory system in Tanzania resulted from advancements in four key areas: 1) The evolution of a legal regulatory framework, 2) Strong stakeholder engagement, 3) Continuous capacity building, and 4) Effective organizational leadership. Tanzania's regulatory system has evolved from being relatively ineffective to leading regional harmonization efforts in East Africa. This progress was not linear, requiring sustained effort, collaboration, and support from key development partners such as the Global Fund, WHO, and UNDP. Future efforts to enhance regulatory effectiveness should focus on creating adaptive systems that respond to changing needs, rather than solely prescriptive functions.

**Funding:** The authors received no specific funding for this work.

**Competing interests:** The authors have declared that no competing interests exist.

## Introduction

Effective regulation of medicines plays a central role in promoting access to good quality and safe medicines and safeguarding the health of millions of people living in low- and middle-income countries [1, 2]. In the absence of effective regulatory systems, the health of the population suffers. In sub-Saharan Africa, sub-standard and falsified medicines comprise 25%–50% of all medicines [3], and cause more than seven hundred thousand deaths annually from Malaria and Tuberculosis alone. In children aged 0–59 months, falsified medicines are estimated to cause more than 120,000 deaths annually [4–6].

In addition, use of sub-standard medications with less than adequate amounts of active ingredients promote the development of resistance, and this is especially important in sub-Saharan Africa where infectious diseases constitute a large portion of the burden of disease. As a result, these medications pose a significant problem for the present, but a far greater problem for the future [7–9].

Substandard and falsified medications also impact negatively on household budgets in sub-Saharan Africa where a large proportion of households lack health insurance and therefore pay for medicines out-of-pocket. In attempts to save money, clients often purchase cheaper medications which are more likely to be falsified or substandard, and this leads to prolonged illness without cure, or development of complications [10]. The economic cost of substandard and falsified human medicines and cosmetics with banned ingredients was recently estimated to represent a relatively large loss of scarce resources for a poor country like Tanzania [11].

Despite the pressing need for robust regulatory frameworks for medicines in sub-Saharan Africa, a combination of systemic challenges, including limited resources, weak governance structures, and inadequate technical capacity, has made it difficult to achieve meaningful progress in this area [12]. Regulators on the continent often face the challenge of attempting to regulate in the face of limited human resource capacity, potentials for regulatory capture, and the difficulty of regulating pharmaceuticals that are mostly produced outside their jurisdiction [13–19]. This is further complicated by the small size of the individual markets they regulate, and the reluctance of large pharmaceutical companies to invest resources to achieve compliance with local regulatory requirements [16, 17]

Active medicine regulation in Tanzania began in 1978, when the pharmaceutical industry was very primitive with only three pharmaceutical manufacturing companies and one college for medical/pharmaceutical trainings. The first public university and first pharmaceutical plant were established in 1961 and 1962 respectively. Today, the Tanzanian pharmaceutical sector is very different from what it was in 1978. There are now about 19 human, veterinary medicines and vaccines plants, 24 medical device facilities, and 10 pharmaceutical and medical device facilities that are under construction, 145 registered importers of medicines [20], 1,600 registered retail pharmacies, 14,000 Accredited Drugs Dispensing Outlets (ADDOs), [21] 49 Pharmacy training institutions (out of which only 4 offer Pharmacy degree courses), and over 500 new graduate Pharmacists registered with the Pharmacy Council annually [22].

To address the challenges of regulating a constantly evolving sector with limited resources, Tanzania's medicine regulatory institutions have had to overcome numerous obstacles under the guidance of various management teams, executive boards, ministerial advisory boards, and evolving organizational structures. They embarked on a series of capacity building interventions that included human resource capacity building, organizational systems strengthening, decentralization of operations, and the promotion of financial sustainability. The TMDA now plays a major role in on-going efforts to harmonize medicines regulation in the East African Community (EAC) and Southern African Development Community (SADC). The progress in the EAC and SADC will have broader impact in other regional economic communities (RECs)

in Africa under the coordination of the African Medicines Regulatory Harmonization (AMRH) Programme [23–29]. Essentially, the AMRH aims to facilitate access to quality, safe and efficacious medicines to the African people by working through the existing political structures, and the regional economic communities (RECs) and eventually operational African Medicines Agency (AMA) [30].

Recently, a consensus has emerged regarding the essential functions necessary for active regulation. These include: 1) Registration of medicines and market authorization, 2) Licensing of key activities like manufacturing, distribution and promotion, 3) Inspections and surveillance, 4) Laboratory access and quality testing, 5) Monitoring of adverse drug reactions, 5) Regulating advertising of medicines, and 6) Promoting rational medicines use [2, 31, 32]. These functions all work together to make available quality assured medicine on the market with the long-term impact of improving population health [31].

The World Health Organization and other international agencies embarked on several systems strengthening programs in low- and middle-income countries with minimal success [13, 32, 33]. In addition to providing technical and financial support for systems strengthening, these agencies have invested significant resources in developing guidelines and frameworks to guide countries on their path towards effective medicines regulation [32, 34].

Over the past several decades, regulation of medicines in Tanzania has undergone an evolutionary process culminating in the establishment of Pharmacy Council for the regulation of professional pharmacy practice and the TMDA for the regulation of medicines, medical devices, and diagnostics see Table 1.

The TMDA is responsible for all regulatory activities related to medicines in Tanzania and it has contributed to Tanzania to become the first in Africa to reach Maturity level 3 of the WHO benchmarking programme [35]. While this is true, much of the improvements in the existence of appropriate structures for a well-functioning of the regulatory system were observed after the establishment of TFDA in midst 2003.

The recorded change in improvement in regulatory system in Tanzania was not a stand-alone effort. Various development partners have contributed both technical assistance (TA) and financial assistance to mention a few; GFATM [36], WHO, MSH [37], Hellen Keller Foundation International, UNDP and University of Gent. Particularly, the GFATM has contributed

**Table 1. Institutions and their regulatory functions in Tanzania, 1978–2020.**

| Regulatory Function | 1978 to 2003 | 2003 to 2019 | After 2019–2020 |
|---|---|---|---|
| Human Medicines regulation | Pharmacy Board | TFDA | TMDA |
| Manufacturing and Importer Premises licensing | Pharmacy Board | TFDA | TMDA |
| Retail and wholesale Premises licensing | Pharmacy Board | Pharmacy Council | Pharmacy Council |
| Registration of pharmacists and pharmaceutical technicians and assistants | Pharmacy Board | Pharmacy Council | Pharmacy Council |
| Food Safety | National Food Control Commission | TFDA | TBS |
| Regulation of cosmetics | Not regulated | TFDA | TBS |
| Medical Devices | Not regulated | TFDA | TMDA |
| In-vitro Diagnostics | PHLB* | TFDA** | TMDA |
| Veterinary medicines | Pharmacy Board | TFDA | TMDA |

*Started 1998

**Started 2017

PHLB: Public Health Laboratory Board

TFDA: Tanzania Food and Drugs Authority

TMDA: Tanzania Medicines and Medical Devices Authority

significantly in Pharmacovigilance systems strengthening, laboratory capacity and post marketing surveillance strengthening while the WHO has continuously provided technical support across key regulatory functions through its Regulatory System Strengthening (RSS) Initiatives [38] and Prequalification Programme [39]. USAID has mainly supported strengthening of laboratory capacity [40].

In-spite of these efforts, no study has been made to systematically understand the impact of these systems strengthening efforts in improving the regulatory capacity and performance in Tanzania. We describe historical evolution of regulation, efforts made, contribution of various players and status/achievement of regulatory system in Tanzania over the period 1978 to 2020. The paper is organized as follows: In the next sections, we present a contextual background of the pharmaceutical sector in Tanzania, then we present the methods used in the study including the analytical framework and data collection approach. This is followed by a presentation of the results from the analysis, discussion of the importance of the findings and then conclusion.

Further, the paper examines the pathway towards success in regulatory systems strengthening in Tanzania from elementary stages to the current system which conforms to the requirements of the WHO maturity level 3 [41] for a stable and integrated regulatory system. The paper narrates evolutionary milestones, successes and demonstrates stepwise approach in regulatory system strengthening in the context of a developing country as lessons worth to be replicated elsewhere.

## Methodology

### Study objectives and analytic framework

The main objective of this study is to assess the evolution of the pharmaceutical regulatory system in Tanzania over the period from 1978 to 2020. Specific objectives of this paper include: To describe the evolution of the national medicines regulatory systems, to document the effects of reforms on medicines regulatory systems in Tanzania, to identify the factors affecting the status of the regulatory system in Tanzania and to describe the achievement of regulatory system in Tanzania over the period from 1978 to 2020.

### Analytical framework

We aligned our approach to the 2019 version of the Global Benchmarking Tool (GBT) terminologies as proposed by the World Health Organization in executing our study objectives [42]. The tool assesses several dimensions of National Medicines Regulatory Systems including National Regulatory Systems, Medicines registration and marketing authorization, vigilance, premises licensing, regulatory inspection, market surveillance and control, clinical trials oversight, laboratory access and testing. Lot release was not included because Tanzania is not a vaccine producing country.

We regrouped these components of the GBT into three major categories to make our analysis more tractable. These are: 1) Regulatory framework, 2) Regulatory capacity, and 3) Regulatory functions (Fig 1).

### Analytical categories

Regulatory framework: This refers to the set of policies, laws and regulations that govern the regulation of medicines. It provides the legal basis for the setup of a regulatory organization and stipulates the main functions and boundaries for regulatory activities including quality and risk management Table 2.

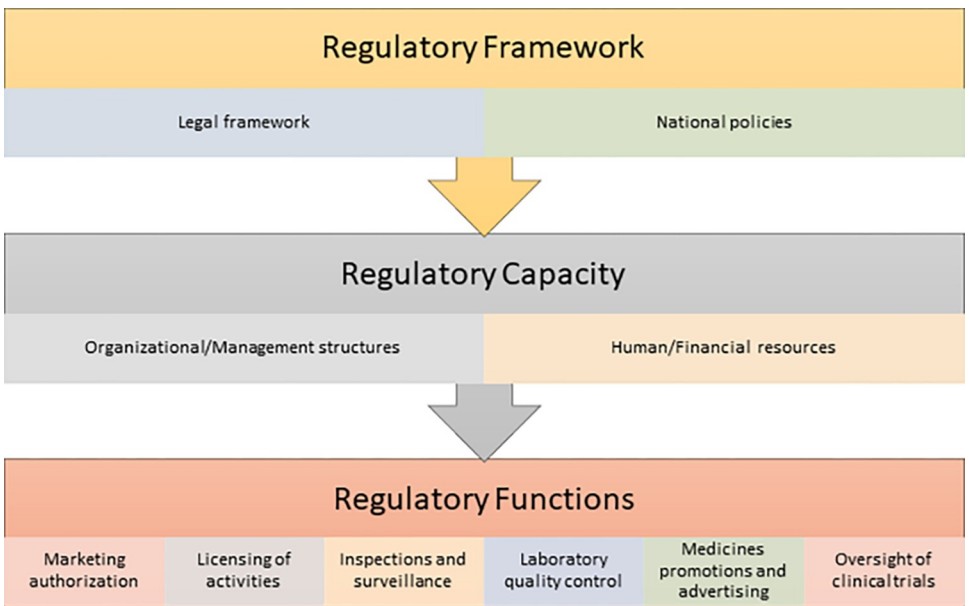

**Fig 1. Analytic framework.**

Regulatory capacity: We defined regulatory capacity to include the physical, organizational, management structures, human and financial resources required for adequate medicines regulation. It also includes Strategic Plan and result monitoring and evaluation **Table 2**.

Regulatory functions: We defined regulatory functions to include the major responsibilities of a medicines regulatory authority as described by WHO [2, 32, 42] These include 1) National Regulatory Systems, 2) Medicines registration and marketing authorization, 3) vigilance, premises licensing, 4) regulatory inspection, 5) market surveillance and control, 6) clinical trials oversight, and 7) laboratory access and testing **Table 2**.

For each analytic component or function, we analysed the status as well as historical trends observed over the period 1978 to 2020. This paper focuses on Tanzania mainland. Details of each analytic category and its importance are described in **Table 2**.

## Data sources and collection

Desk reviews included each Tanzania country's policies, laws, regulations, guidelines, procedures, and Institutional Annual reports. -. The of data included policies (Health Policy, 2007 [43], National Drug Policy 1991 [44]), Various Acts and amendments [45–54]), Regulations (Medicines registration, Orphan medicines, Control of drug promotion, recalling/handling and disposal of unfit products, registration of premises, Pharmacovigilance, Good Manufacturing Practices, Fees and charges, scheduling of medicines and clinical trials control) [55], guidelines (registration of medicines, pharmacovigilance, clinical trials) [56], reports [57] (Ten Years of TFDA 2003–2013, annual reports 2003–2019) [56, 57] peer-reviewed and grey literature, news media archives [58] compared to the 1978 status as baseline.

## Results–Important changes in Tanzania's pharmaceutical sector

Through the assessment, we found documented evidence of improvement in the key medicines' regulatory components over the study period. We present our results in this section while Table 3 provides a summary of our main findings.

**Table 2. Analytical framework–Description of regulatory components and functions.**

| Analytic Categories | Description | Importance |
|---|---|---|
| 1. Medicines Regulatory Framework | Set of policies, laws and regulations of medicines in the country | • Provides the legal basis and mandate for enforcement of regulation.<br>• Defines the boundaries and domains to be regulated.<br>• Defines the form and structure of the institution responsible for medicines regulation.<br>• Provision of fee and charges to facilitate regulatory enforcement. |
| 2. Medicines Regulatory Capacity | Includes the physical, organizational, and management structures required for adequate regulation of medicines | • It's a vehicle for the translation of regulation goals into actual results.<br>• Determines effectiveness and efficiency of regulation.<br>• Establishes, provides and control resources needed for regulation. |
| 3. Medicines Regulatory Functions<br>  a. Medicines registration and marketing authorization | The process of evaluating medicines for quality, safety and efficacy and granting approvals for distribution in the country | • Determines the types and number of medicines that can be legally used in the country. |
| b. Premises licensing | Aims to control participation of important stakeholders in various aspects of the pharmaceutical supply chain. | • Maintains high quality standards by ensuring only qualified premises are allowed for manufacturing, transport, storage and selling of medicines |
| c. Regulatory inspection | Systems and processes that enable continuous monitoring and enforcement of compliance with established standards. | • Establish level of compliance to regulatory standards |
| d. Vigilance | Systems for continuous monitoring of medicines safety | • Monitor, detect, collect, assess, reporting and taking regulatory actions |
| e. Laboratory access and testing | Assessment and continued monitoring of medicines quality using established laboratory methods. | • Provides reliable information on the quality of medicines on the market.<br>• Establish level of compliance to regulatory standards<br>• Fosters evidence-based decision making. |
| f. Market surveillance and control | Regulation of the content, channels, and methods used to disseminate information about medicines, medical devices, and medical products. It also includes Post marketing surveillance of quality of products | • Essential in combating substandard and falsified medicines.<br>• Protects the public from harmful and misleading information about medicines |
| g. Clinical trials oversight | Regulation of all aspects of clinical trials involving new and existing medicines | • Protects the rights of the trial participants.<br>• Ensures that trials follow the appropriate ethical and scientific procedures.<br>• Establish level of compliance to regulatory standards |

## Medicines regulatory framework

The medicines regulatory framework provides the legal basis for the setup of a regulatory organization and stipulates the main functions and boundaries for regulatory activities. It is arguably the most important component of medicines regulation as all other components derive from it. Without a comprehensive regulatory framework, it would be impossible to achieve effective regulation as there will be no legal basis to enforce any regulatory actions. We analysed revisions to the medicine's regulatory framework for Tanzania over the period 1978 to 2020. We compared changes made to prevailing frameworks at different times to the challenges faced within the medicines regulatory space presented in Fig 2.

**Historical context.** The Pharmaceutical and Poisons Act of 1978 [51] when enacted repealed all previous ordinances and established a Pharmacy Board with the responsibility of regulating both pharmacy professionals and pharmaceutical products.

Subsequent changes to the legal regulatory framework occurred in 2002 and 2003 with the passage of the Pharmacy Act of 2002 which established the Pharmacy Council, and the Tanzania Food, Drugs and Cosmetics Act of 2003 which established the TFDA. Both laws were later modified with the passage of the Pharmacy Act of 2011, which effectively repealed and replaced the Pharmacy Act of 2002 and transferred some of the functions of TFDA in the

**Table 3. Overview of key changes in medicines regulatory framework–Summary of findings.**

|  | 1978 to 2002 | 2003 to 2020 |
|---|---|---|
| **Laws/Policies/ Regulations** | • Pharmaceutical and Poisons Act of 1978<br>• Tanzania National Drug Policy (1991)<br>• Pharmacy Act 2002<br>• Specific regulations (poisons list, pharmaceutical and poisons, registration of drug premises, Good Manufacturing Practices, List of notified human drugs, List of notified veterinary drugs, code of conduct for drug promoters Order) | • Tanzania Medicines and Medical Devices Act, Cap 219<br>• Finance Act, No. 8 of 2019<br>• Tanzania Food, Drugs, and Cosmetics Act 2003<br>• Pharmacy Act, Cap 311 National Drug Policy (1991)<br>• National Health Policy (2007)<br>• Regulations (Medicines registration, Orphan medicines, Control of drug promotion, recalling/handling and disposal of unfit products, registration of premises, Pharmacovigilance, Good Manufacturing Practices, Fees and charges, scheduling of medicines and clinical trials control) |
| **Quality and risk management** | Not existed | • Established 2005<br>• ISO9001:2008, certified in 2008 for medicines registration.<br>• Recertified 2012 for Medicines and food regulation<br>Certified ISO9001:2015 in the year 2017 (including risk management)<br>• WHO prequalification of the Tanzania Medicines and Medical Devices (TMDA), medicines laboratory in 2011<br>• Attain ISO17025:2005 in the 2012 |
| **Domains** | Pharmacy Board<br>• Drugs<br>• Pharmacy profession and practice<br>• Food safety regulation transferred to a National Food Control Commission | Pharmacy Council<br>• Pharmacy profession and practice<br>Tanzania Food, Drugs Authority (TFDA)<br>• Medicines<br>• Medical devices<br>• Food safety<br>• Cosmetics<br>TMDA<br>• Medicines<br>• Medical devices<br>• Diagnostics |
| **Challenges** | • Framework did not give regulatory entities sufficient autonomy to function well.<br>• Borderline products which were not fully regulated<br>• Multiplicity of institutions regulating same products | • Overlap of regulatory functions between TFDA, Pharmacy Council and TBS |

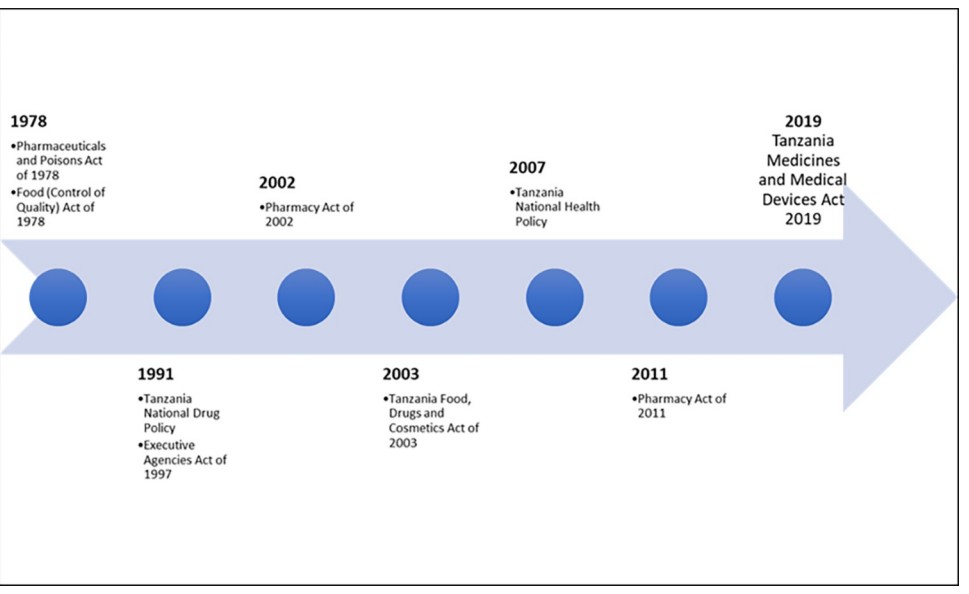

**Fig 2. Changes to Tanzania's regulatory framework 1978 to 2020.**

Tanzania Food, Drugs and Cosmetics Act of 2003 to the Pharmacy Council and later in 2019 TFDA passing two key responsibilities to Tanzania Bureau of Standards (TBS) and become TMDA [59].

**Major challenges.**   In all periods analysed, there were reports of major challenges in medicines regulation that led to clamours for modifications in the prevailing regulatory framework. The most significant challenges faced were related to the separation of the regulatory domains, and in the determination of the form and structure of the institutions saddled with regulatory functions.

In the period between 1978 and 2003, regulation of medicines and pharmaceutical personnel became the responsibility of the Pharmacy Board as stipulated in the Pharmaceutical and Poisons Act of 1978 [51], while regulation of food safety became the responsibility of the National Food Control Commission (NFCC). The NFCC was established under a separate law, the Food (Control of Quality) Act of 1978 [60].

With the passage of both the Pharmacy Act of 2002 [53], and the Tanzania Food, Drugs, and Cosmetics Act of 2003 [52], regulation of medicines and regulation of pharmaceutical professionals were separated. The Pharmacy Council assumed the regulatory responsibility for pharmaceutical personnel while TFDA assumed the regulatory responsibility for pharmaceutical products. In addition, TFDA also assumed the regulatory responsibility for medical devices, foods, and cosmetics. It is not clear what impact these changes in the breadth of regulation and in the assignment of regulatory responsibilities have had on the sector, but they certainly created some discontinuity in regulatory functions during transition periods. During this change, there was a smooth transition since the TFDA served as a caretaker of the Pharmacy Council for a period of one to two years, where two of its staff were delegated to undertake the functions of the Council and allocated funds to support annual operations of the Council. In 2011, the Pharmacy act was revised and this time the regulation of premises i.e. and retail and wholesale outlet was transferred to Pharmacy Council portfolio from TFDA through enactment of Pharmacy act 2011 [54]. However, this transformation elicited a heated debate among key stakeholder questing the capacity of the later in execution of this huge responsibility considering it was under resourced items of human resources, finances and infrastructure required compared to the former institution where the role was initially charged. A recent review on how the pharmaceutical regulation reforms process in Tanzania were conducted which revealed a vested huge, vested interests, displacing a critical analysis of optimal policy options that had a potential to increase efficiency in the regulation of the business of pharmacy [61]. It was noted that reforms moved premises regulation from TFDA which was better placed in terms of infrastructure and skilled human resources. A survey of literature indicated that premises has a direct impact on the quality of medicine than the practice [62, 63] thus it would be a good regulatory practice to charge quality and premises regulation under the same umbrella.

Furthermore, the Public Service Reforms Programme II (PSRP II) [64] implemented by the Government of Tanzania in 1990's envisaged the streamlining of Government departments to improve provision of services to the public. One of the strategies was to establish Government Agencies that would serve customers and the public in a business-oriented manner. As a result of the reforms, the Government enacted the Executive Agencies Act, 1997, as amended in 2009, which enabled some Government departments and institutions to be transformed into Executive Agencies. The Pharmacy Board and NFCC were among institutions which were merged to form TFDA as an Executive Agency under the Ministry responsible for Health. As an executive Agency, the day-to-day operations of TFDA are managed by a Chief Executive Officer while the Ministerial Advisory Board provides guidance on strategic matters affecting the Authority to the Minister of Health.

This governance mechanism provides autonomy for the Chief Executive and his Management Team in running the business of the organization with efficiency and effectiveness. This is contrary to the way the then Pharmacy Board was managed where most of the regulatory decisions including product approvals and licensing of premises were done by the Board on recommendations of various Technical Committees.

**How challenges were addressed.** The most obvious evidence of the system's response to the challenges outlined above is the trail of changes to the laws regulating medicines in Tanzania. These changes, however, represent the culmination of several events that occurred in the intervening periods. They include multiple assessments commissioned to assess, prevailing situations, and multiple discussions and negotiations with stakeholders.

An important event was the development of a national drug policy to serve as a rudder for the entire pharmaceutical sector in Tanzania. The first national drug policy was formulated in 1991 with the release of the Tanzania National Drug Policy of 1991 [44]. It was a key component of the National Health Policy at the time and provided a masterplan for the pharmaceutical sector for the period spanning 1992 to 2000. The policy describes the dissatisfaction of the stakeholders with the existing legal framework (the Pharmaceuticals and Poisons Act of 1978). This dissatisfaction stemmed from the belief of stakeholders that the legal framework was outdated and no longer meeting the needs of the sector. The policy then recommends the replacement of the 1978 act with a new one that enables the Pharmacy Board to "become a properly functioning drug regulatory authority [65].

In 2009, the 1991 national drug policy was reviewed, and a National Medicines Policy was drafted as part of the Health Sector Strategic Plan III [66]. However, this was never approved by the relevant authority pending ongoing review of the health Policy, 2007 which serves as the parent national policy on health.

## Medicines regulatory capacity

We defined regulatory capacity to include organizational structure, administrative and management systems, human and financial resources. In most cases, these derive directly from the legal regulatory frameworks in place at any given period and determines the design of important aspects of the regulatory apparatus including, administration, independence, financial viability, political influence, accountability mechanisms, and transparency. In Tanzania, the effects are manifested in the type of organization saddled with the responsibility for regulation, the powers they enjoy, and their level of independence to perform their duties. We analysed the different regulatory bodies that existed at different times, their roles, and the impact they may have had on medicines regulation in Tanzania. Table 4 provides a summary of our findings.

**Historical context.** Regulation of all activities related to pharmaceuticals in Tanzania was the responsibility of the Pharmacy and Poisons Board in the Department/Ministry of health before 1978. This changed in 1978 when the government transferred regulatory functions to the Pharmacy Board, and the new board bore this responsibility till 2002. In 2002, the regulatory responsibility for pharmaceutical personnel was transferred to the Pharmacy Council while the regulatory responsibility for pharmaceutical products (and other responsibilities) were transferred to TFDA in 2003. These changes to the legal framework had immediate downstream effects on the organizational structure and administrative/management systems adopted to achieve regulatory functions in the country. They also created certain important challenges which we highlight in the next section.

**Major challenges.** The historical separation and combination of regulatory domains at various times in the history of the country was a challenge. Between 1978 and 2002 regulation of food safety was a responsibility under NFCC while regulation of medicines and pharmacy

Table 4. Medicines regulatory capacity–Summary of findings.

| | 1978 to 2002 | 2003 to 2020 |
|---|---|---|
| **Institution** | Pharmacy Board<br>\* National Food Control Commission for food safety | • Pharmacy Council<br>• Tanzania Food, Drugs, and Cosmetics Act 2003 |
| **Organizational management and Level of autonomy** | • Executive Board in the Ministry of Health<br>• Supported by an administrative Secretariat.<br>• Centralized<br>• Headed by a Registrar<br>• Organized in four (4) sections | • Semi-autonomous Agency and Council.<br>• Headed by a Director General supported by six (6) units.<br>• Organized in four (4) directorates.<br>• HQ supported with seven (7) administrative zone offices.<br>• Registrar of the Pharmacy Council reports to the Chief Medical Officer at the Ministry of Health<br>• Director General of Tanzania Medicines and Medical Devices (TMDA) reports to the Permanent Secretary at the Ministry of Health |
| **Resources** | Physical infrastructure<br>• One office building<br>Human<br>• Less than 30 staff.<br>Finance<br>• Government | Physical infrastructure<br>• TFDA/TMDA<br>• Multi-storey Office buildings Dar, Mwanza<br>• Laboratory Buildings Dar and Mwanza<br>Human<br>• More than 200 staff.<br>Finance<br>• Government<br>• Collection of fee and charges<br>• Development partners |
| **Challenges** | Lack of autonomy/independence<br>• Limited human resources<br>• Limited technology<br>• Inadequate financial resources<br>• Lack of organizational capacity building | • Expansion in breath of regulatory domains<br>• Advancement in technology<br>• Low domestic production capacity |

personnel was under the Pharmacy Board. Subsequently, in 2003, regulation of medicines and food safety were again brought together under one organization (TFDA), while the regulation of pharmacy personnel was separated from regulation of medicines and transferred, for the first time, to a separate entity–the Pharmacy Council. Furthermore, TFDA was given added regulatory responsibilities which included regulation of medical devices and cosmetics. These changes created overlapping responsibilities that may have fostered a lack of clarity in the minds of the clients and key stakeholders. The situation got worse in 2010 when the Pharmacy council led policy reforms that sought to expand its' regulatory functions while narrowing the regulatory domains of the TFDA [61].

The third important challenge identified was chronic lack of resources for organizational capacity building. Successive medicines regulatory authorities in Tanzania faced the challenge of lack of resources for organizational capacity building and had to resort to innovative ways of financing such activities. Although regulatory responsibilities were listed in the different ordinances and acts, the resources needed to implement them were not always available creating a gap between intended regulatory results and actual results.

**How challenges were addressed.** The challenges of autonomy were addressed in the batch of act [52] which created semi-autonomous authority–TFDA. TFDA enjoyed reasonable levels of financial independence as most financial decisions related to regulatory activities are made by DG using the funds generated from fees and charges [67]. However, setting of fees and charges for services are not determined by the DG or Registrar. On the other hand, the challenge of separation of domains is currently being addressed through the development of inter-agency coordinating mechanisms. The coordinating mechanisms aim to foster collaboration between these agencies, recommend solutions to overlapping responsibilities, and provide clarity to the public on regulatory responsibilities of each agency.

Challenges of weak organizational capacity was a more difficult challenge to address. In each period, the regulatory authority partnered with domestic and international agencies that provided resources for organizational capacity building interventions. These interventions included strengthening internal management systems, introducing quality management systems, providing staff training opportunities, purchasing modern laboratory equipment, and embarking on infrastructure development projects. The challenge of financial independence and sustainability is partly addressed by setting fees and charges for TFDA under the Tanzania Food, Drugs and Cosmetics Act, 2003 as well as seeking support from development partners, including the Global Fund, USAID, etc. The results have been positive. In the financial year 2003/2004, the annual budget for TFDA was TZS 1.7 billion and this grew to TZS 30 billion in 2015/2016 –an increase of over 1800%. The results from organizational capacity building have also been remarkable. TFDA has attained ISO 9001:2008 certification for the entire organization, ISO/IEC 17025:2005 accreditation and WHO prequalification for its laboratories, and it is currently a lead agency in the medicine's regulatory harmonization programme for the East African Community. The number of staff has increased from 52 in 2003 to 250 in 2016 while operations have expanded from the head office in Dar es Salaam to seven (7) zone offices across the country.

## Medicine's regulatory functions

We defined regulatory functions as per the WHO Global Benchmarking Tool [68] to include: 1) Registration and Marketing Authorization (MA), 2) Vigilance (VL) 3). Market Surveillance and Control (MC) 4. Licensing Establishments (LI), 5) Regulatory Inspection (RI), 6) Laboratory Testing (LT) 7). Clinical Trials Oversight (CT), (See Table 5). For each function, we assessed the historical context, identified major challenges faced and analysed how those challenges were addressed. Due to the evolution of regulatory activities in Tanzania, data from earlier periods were unavailable for some regulatory functions. In such cases, our analysis is restricted to the periods for which data were available.

**Registration and Marketing Authorization (MA).**   Registration of medicines and marketing authorization refers to process of evaluating and approving medicines for sale and use within the country. The process often starts with product assessments and ends with marketing authorization. It is the primary gatekeeping function of medicines regulatory authorities, and its main goal is to ensure that only medicinal products that meet approved standards of quality, safety and efficacy are allowed on the domestic pharmaceutical market.

*Historical context.* Registration and marketing authorization in Tanzania became a legal requirement in 1998 but did not start till the 1999 under the auspices of the Pharmacy Board. Since then, the cumulative number of medicines registered has increased consistently. By the end of 2002, the last full operational year of the Pharmacy Board, two thousand two hundred and twenty-one (2,221) human medicine products were registered for use in Tanzania and these included over 70% of the drugs on the National Essential Drugs List in Tanzania (NEDLIT) [69]. By the end of 2016, the cumulative number of registered medicines had reached nine thousand and eleven (9,011) while the actual number of registered medicines with active registration was two thousand nine hundred and ninety–eight (2,938) human medicines and two hundred ninety—two (292) veterinary medicines.

In addition, the quality of the registration process is more thorough, registration times have reduced from an average of fifteen (15) months to an average of nine (9) months in 2020.

*Major challenges.* At the onset, the Pharmacy Board faced the challenge of introducing a new regulatory requirement in a resource-poor region. Systems had to be developed, staff had to be trained, clients had to be educated, and high standards had to kept. Following the

**Table 5. Medicines regulatory functions: Summary of findings.**

| Regulatory Function | 1978–2002 | 2003–2011 | 2012–2020 |
|---|---|---|---|
| Medicines registration and marketing authorization | Commenced in 1999 under the Pharmacy Board<br>• Gross number of products registered.<br>Challenges:<br>• Difficulties with setting up a new system from scratch: guidelines, funds, qualified staff, technology | • Implemented using non-Comon technical Document (CTD) format.<br>• Mainly paper based submission<br>• Non-web-based Ms-Access registration database<br>• Registration certificates<br>• A total number of 3,554 products were registered.<br>Challenges:<br>• Outdated guidelines<br>• Increased demand for registration<br>• Manually generated certificates.<br>How challenges were addressed:<br>• Adopted WHO recognized standards for registration.<br>• Significantly increased staff capacity building<br>• Hired additional skilled staff for registration | • Implemented using CTD format.<br>• Online web portal for submission of applications<br>• Web-based Regulatory Information Management Information System (IMIS)<br>• Automatic generated registration certificates<br>Challenges:<br>• Increased demand for registration<br>• New molecules for registration versus available expertise<br>• Multi-source products<br>How challenges were addressed:<br>• Recruitment of mixed staff<br>• In-house training programme<br>• Capacity building through participation in WHO prequalification programmes and SWISS Medic Global Health Programme supported by Bill and Melinda Gate Foundation<br>• Adopted WHO recognized standards for registration.<br>• Involvement of external experts |
| Premises licensing | • Regulated manufacturer, wholesale, warehouse and retail premises.<br>Challenges:<br>• Ineffective licensing and effective oversight of premises because of limited human, financial and infrastructure resources<br>How challenges were addressed:<br>• Regulations for registration of premises<br>• Efforts to increase human resources capacity<br>• Expansion of office building for staff | • Regulated manufacturer, wholesale, warehouse and retail premises.<br>Challenges:<br>• Workload<br>• Large geographic distribution of premises<br>How challenges were addressed:<br>• Delegation of power and function to local government<br>• Improved management information systems<br>• Implementation of the Accredited Dispensing Drug Outlet Programme (ADDO) with support from donors such as Global Fund, Management health for Science (MSH) etc.<br>• Organization of TFDA in zone offices<br>• Shifting of mandate on licensing of retails outlets to Pharmacy council | • Regulated manufacturers, importing wholesalers, warehouse and retail premises.<br>Challenges:<br>• Large geographic distribution of premises<br>• Divided mandate on licensing of retail and whole premises<br>How challenges were addressed:<br>• Delegation of power and function to local government<br>• Continuous improvement of management information systems<br>• Guidelines for operation of zone offices |
| Regulatory inspection | • Inspection of foreign and domestic facilities conducted<br>Challenges:<br>• Inadequate resources (staff, equipment, and systems) to support inspection across the country<br>• Workload<br>• Diversity in scope of inspection<br>• Confidentiality<br>How challenges were addressed:<br>• Recruitment of mixed staff<br>• Specialized training | • Inspection of foreign and domestic facilities conducted<br>Challenges:<br>• Challenges from implementing a cost-recovery system<br>• Wide geographic distribution of premises<br>How challenges were addressed:<br>• Increased number of inspectors<br>• Established four (4) zonal offices to facilitate conduction of inspection activities and cost-recovery<br>• Attachment Programme to stringent Authorities | • Inspection of foreign and domestic facilities conducted<br>Challenges:<br>• Challenges from implementing a cost-recovery system<br>• Duplication of inspection activities with the Pharmacy Council<br>How challenges were addressed:<br>• Signing of the MoU between TFDA/TMDA and Pharmacy Council<br>• Introduction of the desk review procedures<br>• Increase of zone offices from four (4) to seven (7) to facilitate conduction of inspection activities and cost-recovery.<br>• Introduce 24 hours inspection at the Port of Entries |

(*Continued*)

**Table 5.** (Continued)

| Regulatory Function | 1978–2002 | 2003–2011 | 2012–2020 |
|---|---|---|---|
| Laboratory access and testing | • There was a laboratory with limited capacity<br>Challenges:<br>• Limited sample testing capacity<br>• Limited scope of laboratory works<br>• Inadequate skill mixed personnel<br>• Insufficient supplies of chemicals, reagents, and consumables<br>How challenges were addressed:<br>• Staff training | • Enactment of the TFDC Act, Cap 219 which established the Tanzania Food and Drugs Authority laboratory.<br>Challenges:<br>• Increased demand for laboratory services<br>• Inadequate laboratory quality management systems<br>How challenges were addressed:<br>• Laboratory building was extended, and new equipment purchased with help from donors<br>• Procurement of chemicals, reference standards and other laboratory consumable with support from the Global Fund<br>• Training of Laboratory staff by support from the Global Fund<br>• Laboratory Quality Management Systems (LQMS) introduced.<br>• Laboratory prequalified as a WHO reference laboratory.<br>• Initiated partnerships for continuous laboratory staff capacity building.<br>• Participation in Proficiency Testing Schemes<br>• The Min-Lab based drug quality assurance programmes | • Expansion of Laboratory services by establishing another Laboratory in Lake Zone located in Mwanza<br>Challenges<br>• Increased demand for laboratory services<br>• Scope of testing that some specialized products cannot be tested<br>• Realization of the cost-recovery<br>How challenges were addressed:<br>• Hiring of more staff<br>• Provide staff with specialized trainings.<br>• Build in a microbiology laboratory with financial assistance from the USAID.<br>• Maintain Laboratory Quality Management Systems (LQMS)<br>• Maintain Laboratory prequalification by the WHO.<br>• Laboratory certified for ISO/IEC 17025:2005<br>• Participation in Proficiency Testing Schemes<br>• Expansion of the Min-Lab testing centers<br>• Procurement of laboratory equipment, chemicals, and consumables through support from the Global Fund |
| Market surveillance and control | Although specified in the regulatory framework, it did not form a major focus of regulatory activities. | Activities that comprise this function are still not a major focus of the regulatory bodies but efforts are underway to change that. | |
| Clinical trials oversight | • Not included in regulatory framework before 2003 | Oversight function given to TFDA with the TFDA Act of 2003.<br>Challenges:<br>• Gaining the acceptance of key stakeholders in the clinical trials space<br>• Lack of capacity to conduct appropriate oversight<br>• Resistance to fees for Good Clinical Practice (GCP) inspections<br>How challenges were addressed:<br>• Creation of a National registry of clinical trials to coordinate efforts.<br>• Establishment of standards for GCP inspections<br>• Staff capacity building | Oversight function continued.<br>Challenges:<br>• Gaining the acceptance of key stakeholders in the clinical trials space<br>• Lack of capacity to conduct appropriate oversight.<br>• Resistance to fees for GCP inspections<br>How challenges were addressed:<br>• Creation of a National registry of clinical trials to coordinate efforts.<br>• Establishment of standards for GCP inspections<br>• Staff capacity building |

transition to the TFDA, the major challenges became that of trying to improve on existing systems, and of meeting globally acceptable standards. At the time, guidelines that were in use were outdated and not able to pass certification standards TFDA was applying for. TFDA also faced a shortage of skilled human resources needed to review application dossiers for quality, safety, and efficacy.

*How challenges were addressed.* TFDA pursued a two-pronged strategy in addressing these challenges 1) It adopted WHO recognized standards, and 2) It embarked on an ambitious programme to improve the number and quality of human resources conducting registration. Guidelines were updated based on existing WHO Prequalification for Medicines Programme and the International Council on Harmonization that led to adoption of a harmonized Common Technical Document (CTD) in 2014 for the East African Community [70]. To improve human resources capacity, training programmes were conducted e.g., hands-on training by WHO Prequalification Programmes in assessment sessions conducted in Copenhagen,

Denmark. The number of staff dedicated to registration was also increased from an average of five (5) in 2003 to fifteen (15) in 2016. Registration staff are further supported by a pool of part-time assessors who are called-upon intermittently to expedite assessments and approvals. A more recent development was the launching of an initiative to address regional skills of assessors through a consortium between TFDA and School of Pharmacy Muhimbili University of Health and Allied Sciences (MUHAS). The TFDA/MUHAS consortium (TFDA-MUHAS RCORE) has been designated a Regional Centre of Regulatory Excellence in Medicines Evaluation and Registration [71]. The primary goal of the Regional Centre of Regulatory Excellence in Medicines Evaluation and Registration is to assist National Medicines Regulatory Authorities (NMRAs) in the region to build up national and regional capacity in pre-approval scientific evaluation of medicines so that the public can access these medicines and be assured that they meet acceptable standards of quality, safety, and efficacy. This ensures TFDA continues to improve its work on effective medicines registration.

**Vigilance (VL).** Vigilance includes all activities carried out to continuously monitor the safety, and compliance with the established guidelines and regulations. They encompass monitoring of Adverse Drug Reaction (ADR), monitoring of Adverse Events Following Immunization (AEFIs) and monitoring of events and incidents due to medical devices and diagnostics.

*Historical context*. Formal vigilance activities started later with the introduction of a Spontaneous Pharmacovigilance System in 1993 [72, 73] for the monitoring of adverse drug reactions and vaccine safety in the country. The vigilance of events related to medical devices and diagnostics was introduced after commencement of regulation of medical devices and diagnostics by the TFDA in 2014. The yellow form which is considered as the main tool for collection of ADRs and AEFIs was introduced in the year 1989 [72, 74] prior to formal pharmacovigilance activities. The first draft of Pharmacovigilance guidelines which stipulated minimum requirements and capacity for Pharmacovigilance by key stakeholders like healthcare workers, patients and Marketing Authorization Holders was first developed in 2009. Publication of pharmacovigilance regulation in 2018 [75] brought more supervisory muscles to the National Regulatory Authority to enforce the responsibilities of all stakeholders in continued monitoring of quality and safety of medicines.[76, 77].

*Major challenges*. The role of reporting suspected ADR by all stakeholders was not smoothly comprehended clarify/expand/low reporting that led into inability of the TFDA to make evidence-based decisions on safety of medicines and vaccines due to few reports of ADR and AEFI. There was not an established system for collection and management of events and incidents due to medical devices and diagnostics [76].

*How challenges were addressed*. The challenges were addressed by conducting regular training and sensitization activities to health care workers and other Pharmacovigilance stakeholders on reporting of ADRs and AEFI. The TFDA improved the connection to various Pharmacovigilance stakeholders and PV activities started being incorporated with Public Health Programme like the National Malaria Control Programme (NMCP). The guidelines for surveillance of AEFI were developed by the TFDA and Expanded program on Immunization (EPI) jointly which stipulated the channels of reporting AEFI from the point of identification at possible low level to the National Centre, TFDA. In addition, the regulations for Pharmacovigilance brought mandatory reporting of ADR by all PV stakeholders and introduced the requirement of Qualified Person for Pharmacovigilance which is key for Marketing Authorization Holders [78]. TMDA has also introduced an electronic system [79] for reporting adverse drug reactions in Tanzanian which make it easier to communicate with VigiFlow [77]. The programme allows data to be transferred into the WHO International Database of suspected adverse drug reactions, VigiBase, by uploading the XML file in the user's VigiFlow webpage. The software acts as a local database and shortens the time for data entry into VigiFlow [80].

**Market surveillance and control MC.** Market surveillance and control include all activities carried out to continuously monitor the safety, efficacy, and compliance with the established guidelines throughout the supply chain. They include systems and processes that enable the regulatory authority to monitor and enforce continuous compliance with established standards of Good Storage and Distribution Practices (GSDP), systems to support post-market assessments of medicines quality and control of product promotion and advertisement.

Medicines promotion and advertising refers to systems setup to ensure compliance with existing laws and regulations affecting marketing of medicines and medical devices. This regulatory function is very important because it ensures that information reaching the public meets the highest possible standard and that the public is not misled by exaggerated information or deceptive advertising.

*Historical context.* Formal inspection of products in the supply chain started by the Pharmacy Board in 1992 with the goal of ensuring that stakeholders comply with GSDP regulations [81]. To strengthen this a regulation for the Recall, Handling and Disposal of Unfit Medicines and Cosmetics) Regulations, 2015 was developed [82]. Activities in pursuit of these goals were carried out both domestically, and internationally for medicines destined for Tanzania.

Formal surveillance activities started later with the introduction of a Post-Marketing Surveillance (PMS) programme in 2009 [75] for continued monitoring of quality of medicines from the National Essential Medicines List, and later on expanded to cover an entire range of products authorized for use and circulating on the market in Tanzania.

We found no significant changes in the functions related to regulation of promotion and advertising of medicines over the period studied. In all periods analysed, the prevailing legal frameworks provided for regulation of advertising and promotion, and specific regulatory guidance was provided under the different Acts [51, 83]. However, the means of enforcement varied over time, reflecting the changing forms of media through which medicines are promoted or advertised rather than a change in the scope or form of regulation.

*Major challenges.* Market surveillance activities under the Pharmacy Board (1992 to 2003) faced two major challenges, namely: lack of adequate laboratory support, and inadequate resources (staff, equipment, and systems) to support surveillance activities across the country. Another challenge was lack of tools and methodologies for processing of applications for importation of the products and tracking of product batches throughout the supply chain.

The major challenge in this area has been publicity of medicines adverts and promotional materials that are misleading. Previous Acts before 2003 had provisions for control of medicines promotion and advertising but enforcement become evident after enactment of Tanzania Food, Drugs and Cosmetics Act, 2003 [83] and setting up of guideline for control of promotion [84]. Not much was put for public education programmes. Sales and advertisement of medicines and cosmetics in publics transport. Science and technological advancement that resulted in wide promotion of products in social media [85–87]. Mushrooming of private commercial medial houses resulting in regulatory oversight overload.

*How challenges were addressed.* To address the challenge of inadequate resources (staff, equipment, and systems), both the Pharmacy Board and TFDA partnered with stakeholders to implement several systems strengthening programs. These include the implementation of the Quality Assurance Programme which included a big drive to strengthen inspections at ports of entry established in 2002.

TFDA also decentralized its activities by strengthening the capacity of its zone offices to implement market surveillance activities aimed at identifying substandard and falsified medical products throughout the supply chain. In this effort TFDA started establishing Zone offices located in major cities and regions and serving other neighbouring regions. By the year 2019

TFDA had established a total of 8 such offices throughout the country, this brought market surveillance activities closer to the public and especially in hard-to-reach parts of the country.

TFDA also embarked into ambitious efforts which saw significant expansion of its main laboratory in Dar es Salaam as well as building of an ultra-modern state of the art quality control laboratory in the Lake Zone area with support from GF. This further strengthened laboratory testing capacity of the Authority and facilitated timely analysis of product samples collected from as many parts of the country as possible.

The Tanzania Food, Drugs and Cosmetics Act, 2003 provided for control of medicines promotion and advertising. To facilitate enforcement of the law, regulations were issued (The Tanzania Food, Drugs and Cosmetics (Control of Drugs and Herbal Drugs Promotion) 2010 [88] and approval process for promotional materials was set up. Public education programmes were enhanced with establishment of a dedicated unit within TFDA. The units run several public educational programs on TV and radio to provide balanced information to the public.

**Licensing Establishments (LI).**   Licensing is an important regulatory function that allows participation in the pharmaceutical sector. It gives the regulator the ability to control participation of important stakeholders in various aspects of the pharmaceutical supply chain. It includes licensing of manufacturing, and all aspects of distribution including import/export, wholesale, and retail. By performing this function effectively, a medicines regulatory authority can maintain high levels of quality in the system by ensuring that only parties who have met required standards are licensed to operate.

*Historical context.* Licensing has been performed by all regulatory authorities in Tanzania since even before 1978. The Pharmacy Board (1978–2003) was responsible for the licensing of activities related to production, distribution (including sale) and use of pharmaceutical products. This changed in 2003 when the Pharmacy Council assumed the role of regulating pharmaceutical professionals while TFDA assumed the role of regulating pharmaceutical products and premises involved with manufacture, storage and distribution of such products.

In addition, an important expansion of the domains for licensing was included in the Tanzania Foods, Drugs, and Cosmetics Act of 2003. In this Act, TFDA was required to license activities related to medical devices, cosmetics, foods, and other herbal medicines. This expanded the powers of the Authority to effectively regulate other products that have a direct effect on health of individuals [89–92].

*Major challenges.* Some major challenges faced by the different authorities over the period studied include problems with licensing retail outlets, regulatory functions that overlap with other agencies, insufficient cost-recovery mechanisms, and weak internal communication and management systems for licensing. The licensing process also involved inefficient processes and multiple meetings by licensing committees before a decision is reached by the Board.

As the market for medicines grew in Tanzania, there was an increase in the number and variety of small-scale retail outlets for medicines called '*Duka la Dawa Baridi'* (DLDB) [89, 93, 94]. Many of these outlets carried a mix of traditional and orthodox medications and were not permitted to stock more complex medicines also known as "Part One" (Prescription Only) medicines. This restriction did not stop some from retailing in these medicines and due to lack of resources it was difficult to regulate these DLDBs [95, 96].

The prevailing lack of resources was exacerbated by the absence of reliable cost-recovery mechanisms that reflect the real cost of doing business. Although authorities in each period charged fees, the fees were low and not directly managed by the authority thereby creating limitations with financial planning and implementation.

More recently, the overlap of regulatory functions between the Pharmacy Council and TFDA created difficulties in regulation [97] while the increase in volume of medicines imports and complexity of the market created challenges with maintaining reliable communication

mechanisms to convey up-to-date information on the status of licenses issued, suspended, or withdrawn.

*How challenges were addressed.* Under the aegis of the Pharmacy Board, ADDO [89, 93] programme was introduced to address the challenge with licensing and regulating DLDBs. Through these programmes, the Pharmacy Board partnered with organizations like Management Sciences for Health, DANIDA and The Global Fund to provide training, accreditation, and supervisory support to transform existing DLDBs into ADDOs. The programme is now managed by the Pharmacy Council and its success has been widely reported [89, 93].

Furthermore, with the autonomy enjoyed by TFDA, it was able to increase the fees it charged for its services. It was able to charge fees that more closely match the cost of conducting its licensing functions, ranging from import/export licenses to premise licenses. All the fees help offset the cost of providing services and makes the Authority less reliant on the Government Subvention. TFDA also leveraged its internal electronic management information system to facilitate communication of the status of licenses between the central office, zonal offices, and ports of entry teams. The committee meetings step in processing of licences was abolished and decisions on licensing of premises even further delegated to TFDA Zone Managers.

**Regulatory Inspections (RI).** Inspections and surveillance include all activities carried out to continuously monitor the safety, quality, and compliance with the established guidelines. They include systems and processes that enable the regulatory authority to monitor and enforce continuous compliance with established standards–e.g. assessments of compliance with Good Manufacturing Practices (GMP) and/or Good Distribution Practices (GDP), systems to support post-market assessments of medicines quality, and monitoring of adverse drug reactions

*Historical context.* Formal inspection of premises was started by the Pharmacy Board in 1992 with the goal of ensuring that stakeholders comply with GMP [98], GDP and Good Storage Practices (GSP) [81], and a regulation for the Recall, Handling and Disposal of Unfit Medicines and Cosmetics) Regulations, 2015 [82] Activities in pursuit of these goals were carried out both domestically, and internationally for medicines destined for Tanzania.

Formal surveillance activities started later with the introduction of a Spontaneous Pharmacovigilance System in 1993 for the monitoring of adverse drug reactions in the country, followed by a PMS programme in 2009 and publishing of pharmacovigilance regulation in 2018 [75] for continued monitoring of quality of commonly used medicines from the National Essential Medicines List, and later, the introduction of an Electronic Adverse Drug Monitoring System in 2016.

*Major challenges.* Regulatory Inspections and surveillance activities under the Pharmacy Board (1992 to 2003) faced two major challenges, namely: lack of adequate laboratory support, and inadequate resources (staff, equipment, and systems) to support inspection and surveillance activities across the country. Following the enactment of the TFDA Act, and the splitting of the regulatory functions between the Pharmacy Council and TFDA, a third challenge emerged: overlap of activities which created some confusion among clients about the responsibilities of the different regulatory authorities. Under the new system, TFDA was responsible for regulating medical products while the Pharmacy council regulated pharmacy personnel. Confusions arose because the lines were not clear as regards inspection and surveillance activities. Finally, TFDA also faced the challenge of cost-recovery from implementing its inspections and surveillance functions.

*How challenges were addressed.* To address the challenge of inadequate resources (staff, equipment, and systems), both the Pharmacy Board and TFDA partnered with stakeholders to implement several systems strengthening exercises. These included the implementation of the

Drug Quality Assurance Programme which included a big drive to strengthen inspections at ports of entry, and the implementation of the ADDO programme which included a significant investment in scaling up inspections of ADDO premises and activities nationwide.

TFDA also decentralized its activities by strengthening the capacity of its zone offices to implement inspection and surveillance activities. The introduction of the Pharmacovigilance system and PMS system created opportunities to improve coordination of activities between the zone offices and the centre, leading to improvements in service delivery.

The passage of the Pharmacy Act of 2011 attempted to resolve the challenge of overlapping functions. This Act transferred the powers to licensing domestic premises (wholesale, retail and dispensing outlets) to the Pharmacy Council while TFDA retained the powers to continue inspection of products throughout the supply chain and licensing of manufacturers and wholesalers involved with importation. Although this provides some clarity at the level of the regulators, it is still confusing for clients and both regulatory authorities are currently developing formal coordination mechanisms to improve efficiencies.

Lastly, to address the challenge of inadequate cost-recovery mechanisms, TFDA revised its schedule of fees and charges in 2005 through the Tanzania Food, Drugs and Cosmetics (Fees and Charges Regulations (The TMDA), as amended in 2011 and 2015 [67]. Under this system, fees are charged for registration of products, import fees at 2% Freight on Board on the value of imported medicines is assessed as a fee to cover inspection and surveillance activities related to that consignment of medicines. In addition, clients pay for licenses and meet the cost of inspections to verify compliance with good manufacturing practices.

**Laboratory testing.** We defined laboratory testing as the ability of the medicines regulatory authority to objectively measure the quality of medicines or medical devices it regulates. This function requires a fully functional laboratory with appropriately installed quality management systems (QMS) and good laboratory practices. It is important because it enables the regulatory authority to make evidence-based decisions following assessments of quality of medicines and rely less on subjective measures such as inspectors' opinions.

*Historical context.* The quality control laboratory in Tanzania was established in 2000 by the Pharmacy Board as part of its Quality Assurance Program. Since then, the laboratory has steadily increased its capacity to conduct more complex tests and handle larger volumes reducing sample backlogs. The TFDA piloted the use of Minilab kits, a thin-layer-chromatographic based drug quality testing technique, in a two-tier quality assurance program [99]. The program was intended to improve testing capacity with timely screening of the quality of medicines as they enter the market. Initially six mini laboratories were established in six regions and has grown to 17 minilab centres by 2019 to serve as remote facilities for preliminary field screening test. The number of medicines analysed has risen from an annual average of two hundred and ninety-two (292) medicines in 2002 to two thousand one hundred and fourteen (2358 in 2019/2020 [57]. In 2016, laboratory information management system [100] was launched to ensure efficiency of laboratory services.

*Major challenges.* Upon creation of the national laboratory, it faced one major challenge: its lack of sufficient capacity to meet the increasing demands of the regulatory system. It faced this capacity deficiencies in three important areas namely: 1) Insufficient laboratory equipment both types and number and supplies, 2) Insufficiently trained laboratory staff, and 3) Inadequate laboratory quality management system.

*How challenges were addressed.* To address these challenges, the Pharmacy Board before, and its successor TFDA, embarked on a series of capacity building interventions with the support of several international agencies including the Global Fund to fight AIDS, Tuberculosis and Malaria, the United States Agency for International Development and the World Health Organization [101, 102].

Consequently, support was received from these organizations in the form of modern laboratory equipment, financial support for laboratory supplies, sponsored opportunities for staff development in local and international fora, and technical support to improve internal laboratory processes. In addition, TFDA embarked upon a massive expansion of the laboratory facilities with government funding, adjusted prices to improve cost recovery, and introduced a Laboratory Quality Management System (LQMS) to improve quality assurance in the laboratory.

As a result of these efforts, the TFDA laboratory became prequalified by the WHO as an international reference laboratory in 2011 [103] and received the ISO/IEC 17025:2005 [102] accreditation in 2012 for good laboratory practices.

**Medicines promotion and advertising.**   Medicines promotion and advertising refers to systems setup to ensure compliance with existing laws and regulations affecting marketing of medicines and medical devices. This regulatory function is very important because it ensures that information reaching the public meets the highest possible standard and that the public is not misled by dangerous information or deceptive advertising.

*Historical context*. We found no significant changes in the functions related to regulation of promotion and advertising of medicines over the period studied. In all periods analysed, the prevailing legal frameworks provided for regulation of advertising and promotion, and specific regulatory guidance was provided under the different Acts [51, 83]. However, the means of enforcement varied over time, reflecting the changing forms of media through which medicines are promoted or advertised rather than a change in the scope or form of regulation.

*Major challenges*. The major challenge in this area has been publicity of medicines adverts and promotional materials that are misleading. Previous Acts before 2003 had provisions for control of medicines promotion and advertising but enforcement become evident after enactment of Tanzania Food, Drugs and Cosmetics Act, 2003 [83] and setting up of guideline for control of promotion [84]. Not much was put for public education programmes. Sales and advertisement of medicines and cosmetics in publics transport. Science and technological advancement that resulted in wide promotion of products in social media [85–87]. Mushrooming of private commercial medial houses resulting in regulatory oversight overload.

*How challenges were addressed*. The Tanzania Food, Drugs and Cosmetics Act, 2003 provided for control of medicines promotion and advertising. To facilitate enforcement of the law, regulations were issued (The Tanzania Food, Drugs and Cosmetics (Control of Drugs and Herbal Drugs Promotion) 2010 [88] and approval process for promotional materials was set up. Public education programmes were enhanced with establishment of a dedicated unit within TFDA. The units run several public educational programs on TV and radio to provide balanced information to the public.

**Clinical trials oversight (CT).**   This regulatory function refers to the responsibility of the regulatory authority to ensure that the conduct of clinical trials follows stipulated best practices, and that the population is protected from unnecessary study-related harm. By playing this role, regulatory authorities are able to ensure that control the quality of clinical trials conducted within its borders.

*Historical context*. Before 2003, there were no laws formally requiring the regulation of clinical trials in Tanzania. The only documented oversight function by the Pharmacy Board was limited to the registration of the vaccine, and monitoring for adverse reactions. With the passage of the Tanzania Food, Drugs and Cosmetics Act 2003, the powers to regulate clinical trials were given to TFDA. It was therefore legally mandated that all clinical trials required a written authorization from the TFDA prior to commencement.

*Major challenges*. TFDA faced two major challenges in its efforts to regulate clinical trials. Its biggest challenge was in gaining the acceptance of key stakeholders in the clinical trials

space. Some queried the wisdom in saddling TFDA with the responsibility of overseeing clinical trials and therefore resisted TFDA's initial attempts at providing oversight. The situation was made worse when TFDA introduced fees for Good Clinical Practice (GCP) inspections. The second biggest challenge was the lack of trained staff at TFDA to conduct GCP inspections or provide adequate oversight. These challenges conspire to make the job of providing oversight for clinical trials very difficult for the newly created TFDA.

*How challenges were addressed.* To address the perceived inability of TFDA to provide good oversight for clinical trials, the authority immediately initiated several measures to bolster its capacity and reputation with the stakeholders. Since then, TFDA has released extensive regulatory guidelines for clinical trials [104] and associated regulations [105], which among other things provides for 1) The creation of a National Registry for clinical trials [106], 2) Creation of data and safety monitoring committees, 3) Establishment of standards for monitoring of good clinical practices, and 4) requirements for adequate insurance/indemnity cover for trial participants. TFDA also created a special unit dedicated to these activities.

TFDA also developed a menu of short- and long-term training programs for its staff to acquire new knowledge and skills on regulation of clinical trials. Some are conducted as in-house training facilitated by trained staff and external experts on clinical trials, while others occur outside TFDA. Outside training includes attachments to other institutions for hands-on training, or formal courses in academic/training institutions.

## Discussion: Factors contributing to improved medical product regulation

From the foregoing, the system to regulate medicines in Tanzania has evolved over time. We see progressive change, beginning from the era of the Pharmacy Board (1978 to 2003) to the current operations of the Pharmacy Council (2002 to date) and the Tanzania Food and Drugs Authority (2003 to 2019) and to current TMDA (2019 to date).

While it is impossible to trace any direct causal-effect links between the events highlighted above, some careful analysis highlighted four important factors that may have contributed to the progressive changes seen. They include: 1) An evolving legal regulatory framework, 2) Sustained efforts at systems strengthening and capacity building, 3) A history of effective organizational leadership, and 4) An actively engaged community of stakeholders. We address each of these factors below:

### Evolving legal regulatory frameworks

Between 1978 to 2020, four major reforms to the legal and regulatory frameworks for medicines have been made. This fact alone, suggests a culture of continuous quality improvement in the legal framework to address prevailing concerns of stakeholders.

The 1978 Pharmaceuticals and Poisons Act created a Pharmacy Board and transferred all powers to regulate medicines and pharmacy profession to it while the Food (Control of Quality) Act of 1978 [60] established a NFCC under the Ministry of Health and transferred all activities related to food regulation to it. Medicines regulation was now under one functional organization which improved coordination of activities. However, as time passed, it became apparent that for further improvements in regulation to be pursued, a comprehensive regulatory authority needed to be established which will operate as an autonomous agency under the Ministry of Health. These sentiments were clearly articulated in the National Drug Policy of 1991 [44].

The Tanzania Food, Drugs, and Cosmetics Act, 2003 and the Pharmacy Act 2002 both addressed the concerns of autonomy. Both Acts created new organizations that were self-

governing, free from undue political influence from the parent ministry, and able to exercise a reasonably wide degree of financial autonomy. The Acts also re-introduced fragmentation of regulatory activities between the Pharmacy Council (pharmacy profession practice regulation), and TFDA (medical products regulation). The TFDA Act also re-integrated food regulation with medicines regulation under one organization

Further reforms were initiated by the Pharmacy Council and led to the enactment of the Pharmacy Act of 2011 which further transferred some responsibilities from TFDA to the Pharmacy Council. The trend seems to suggest a mix regulation of medicines and pharmaceutical professionals a move that elicited a very heated debate among key stakeholders. This prompted a study to investigate the how the process was conducted and published the findings two years later 2013 with recommendations which were not in favour of the reform especially transfer of premise regulation to be combined with profession and not with product quality [61]. While it is not clear what direction this will ultimately take, the process of updating the legal framework to reflect stakeholder and sectoral needs is commendable. But this process needs to be managed carefully in order not to introduce instability in the sector that might arise from frequent changes.

## Sustained systems strengthening and organizational capacity building

We found many documented examples of systems strengthening and organizational capacity building efforts that occurred over the forty-two-years period between 1978 and 2020.

Between 1978 and 2003, the Pharmacy Board conducted several initiatives aimed at improving the medical product regulation through systems strengthening. Most notable, were attempts to build and commissioned a quality control laboratory in 2000, initiated the Drug Quality Assurance Programme [107] and the Accredited Drug Dispensing Outlets programme [92] in 2002, and released several regulatory guidelines. In addition, it partnered with the government and several donor organizations to support training of key personnel in domestic and international institutions, and in the procurement of laboratory equipment.

With the creation of TFDA, the focus shifted slightly towards strengthening management systems. The newly formed organization quickly developed the first of three-successive five-year strategic plans to guide its annual business plans, introduced an organization-wide Quality Management System, and a Laboratory Quality Management System [108]. It also commenced efforts to move itself in the direction of financial autonomy with the revision of fees and charges from time to time [107]. In this period, TFDA achieved ISO 9001:2008 certification [109] and ISO IEC 17025:2005 [102] accreditation as well as WHO pre-qualification [103] of its medicine quality control laboratory as an international reference laboratory and implemented an organization-wide electronic management information system. It has also successfully introduced several programmes such as the Post-Marketing Surveillance system, the electronic Adverse Drug Monitoring system, and a clinical trials oversight regulatory system. It has also continued to support individual technical capacity development of staff by organizing internal trainings or sponsoring domestic and international training based on annual training programmes for its staff.

Throughout the period, there is evidence of attempts to continuously improve the system. For example, at the onset, TFDA was organized into functional directorates based on their core functions e.g. Product Evaluation and Registration, Inspections and Surveillance and Laboratory Services. However, with the review of the first strategic plan it was observed that this arrangement was hampering effectiveness, and the second strategic plan proposed a change to the current structure of organization by product i.e. Medicines, Food Safety and Laboratory Services). Systems like these and others, create atmospheres of continuous quality improvement, and over time, lead to remarkable improvements.

### History of continuity and effective organizational leadership

Closely related to the point above is the fact that between successive periods, there was considerable continuity in organizational leadership and programmes. For example, the ADDO programme was initiated by the Pharmacy Board and piloted and rolled out by TFDA and now managed by the Pharmacy Council, The Standard Operating Procedures and Guidelines were introduced by the Pharmacy Board but were expanded and institutionalized by TFDA. Other examples include the commissioning of a laboratory by the Pharmacy Board and its expansion and accreditation under TFDA, or the initiation of training programmes under the Pharmacy Board and its expansion and formalization under TFDA. There has also been remarkable continuity of organizational leadership with smooth transitions since the establishment of TFDA from 2003 to date.

Although these observations may seem unimportant, they become significantly more important when viewed considering the remarkable discontinuity observed in countries (or sectors) when laws and programmes changed.

One explanation for this continuity is the transition of personnel across organizations. Most of the founding staff of TFDA and the Pharmacy Council were inherited from the defunct Pharmacy Board and NFCC. Many of those initial staff have grown through the ranks and over the years have occupied key leadership positions in both TFDA and Pharmacy Council. It is also important to note that these leaders were also key stakeholders who fought for the creation of TFDA and Pharmacy Council, so they were very aware of the challenges they faced at the time, unified on the approaches to solve those challenges.

### Actively engaged community of stakeholders and international donors

No success is possible without the active participation of important stakeholders. This also applies to the regulation of medicines in Tanzania. In all aspects reviewed in this study, there was clear documentation of active participation of key stakeholders such as parliamentarians, government employees, pharmaceutical professionals, manufacturers, importers, distributors, and users of pharmaceuticals (Hospitals, Clinics, Health centres, Dispensary, Community pharmacy and ADDO shops).

Successful changes to the laws governing medicines regulation required prolonged discussions among stakeholders in critical sub-sectors. This was evident in the enactment of the 1978 [51] Acts. An interesting development between the 1978 and 2002/2003 Acts was the implementation of various assessment of the pharmaceutical sector and subsequent development of a national drug policy that included participants outside the medicines regulatory sector. This drug policy and its accompanying masterplan for pharmaceutical sector development in Tanzania, led to the creation of TFDA and the Pharmacy Council.

International donors form an important constituent of stakeholders that have made significant contributions towards strengthening regulatory systems in Tanzania. The most significant contributions made by these stakeholders include provision of technical support, financing of equipment and infrastructural development, and financing of individual capacity building efforts. They also funded/supported several landmark initiatives in Tanzania such as the ADDO programme (USAID, MSH, The Global Fund Malaria Programme), development of the Tanzania medicines policies in 1991 (WHO), funding successive health systems strengthening grants [36], and strengthening of pharmaceutical procurement and distribution system (GFATM) [36]. Between 2007 and 2016, international donors have contributed over US $15 billion. In recent years, GFATM has become the biggest contributor to TFDA's systems strengthening exercises. Between 2007 and 2016 it has contributed US$ 7,479,025.93 towards TFDA projects including pharmacovigilance, post-marketing surveys, laboratory capacity

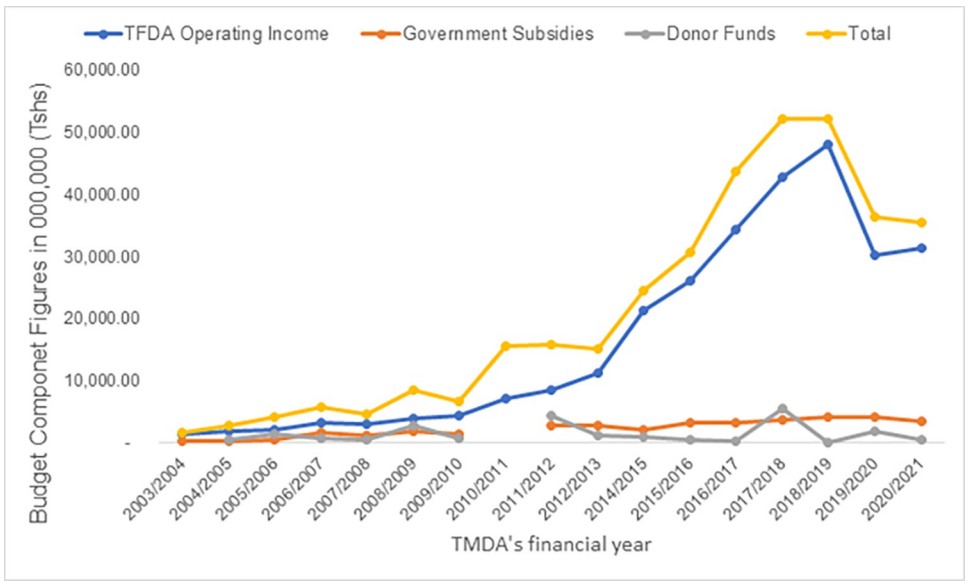

**Fig 3. TMDA Annual Income 2003 to 2020.**

building, and implementation of the ADDO programme. Fig 3 and Table 6 shows the various financial strands and their contributions 2003 to 2020. TMDA's internally generated revenue and government subsidies increased steadily since the merge of Pharmacy Board and TFC in 2003 through enactment of Tanzania Food and Drug Authority [52]. This budget decreased by almost 37% in the internally generated funds and 30% of the overall budget by 2019/2020. This decreases coincided with a major regulatory reform with Finance Bill 2019 [110] which shifted food and cosmetic regulation to TBS.

## Conclusions

The evolution of Tanzania's medicines regulatory system is remarkable, transitioning from a relatively ineffective system to one that currently leads regional regulatory harmonization efforts in the East African region. However, our analysis shows that these improvements did not follow a linear path. Instead, they required long-term efforts to strengthen regulatory systems, achieved through successive changes to legal regulatory frameworks, active participation of stakeholders, and effective organizational leadership within regulatory institutions.

Development partners have played a significant role in the successful journey that the TMDA takes pride in today. Some noteworthy partners include the Global Fund to Fight

**Table 6. TFDA breakdown of donor support (2007 to 2020).**

| Donor | Total Support (2007 to 2020) |
|---|---|
| Global Fund to Fight AIDS Tuberculosis and Malaria (GFATM) | $ 7,479,025.93 |
| World Health Organization (WHO) | $731,275,120.00 |
| Management Sciences for Health (MSH) | $8,327,733.00 |
| Hellen Keller Foundation International | $125,054,323.00 |
| United Nations Development Programme (UNDP) | $242,536,000.00 |
| University of Gent | $618,103,527.00 |
| Others | $ 2,334,793,558.41 |
| Total | $ 7,760,090,261.4 |

AIDS, Tuberculosis, and Malaria (GFATM), the World Health Organization (WHO), the Management Sciences for Health (MSH), the Helen Keller Foundation International, the United Nations Development Programme (UNDP), the University of Gent, and others.

Considering this, attempts to enhance medical product regulation may be more fruitful by shifting the focus away from solely prescriptive regulatory functions. Instead, the emphasis should be on fostering conditions that enable stakeholders to build regulatory systems that can adapt to their changing needs.

## Author Contributions

**Conceptualization:** Adam M. Fimbo, Hiiti B. Sillo, Alex Nkayamba, Sunday Kisoma, Yonah Hebron Mwalwisi, Eliangiringa Kaale.

**Data curation:** Adam M. Fimbo, Alex Nkayamba, Sunday Kisoma, Sarah Asiimwe.

**Formal analysis:** Alex Nkayamba, Sunday Kisoma, Sarah Asiimwe, Patrick Githendu, Linden Morrison.

**Funding acquisition:** Rafiu Idris, Osondu Ogbuoji, Linden Morrison, Jesse B. Bump.

**Investigation:** Adam M. Fimbo, Hiiti B. Sillo, Sunday Kisoma, Yonah Hebron Mwalwisi.

**Methodology:** Adam M. Fimbo, Hiiti B. Sillo, Alex Nkayamba, Sunday Kisoma, Yonah Hebron Mwalwisi, Osondu Ogbuoji, Eliangiringa Kaale.

**Project administration:** Adam M. Fimbo, Hiiti B. Sillo, Rafiu Idris, Linden Morrison.

**Resources:** Adam M. Fimbo, Hiiti B. Sillo.

**Supervision:** Linden Morrison, Eliangiringa Kaale.

**Validation:** Adam M. Fimbo, Eliangiringa Kaale.

**Visualization:** Adam M. Fimbo, Osondu Ogbuoji.

**Writing – original draft:** Adam M. Fimbo, Hiiti B. Sillo, Alex Nkayamba, Sunday Kisoma, Yonah Hebron Mwalwisi, Sarah Asiimwe, Osondu Ogbuoji, Eliangiringa Kaale.

**Writing – review & editing:** Rafiu Idris, Patrick Githendu, Linden Morrison, Jesse B. Bump, Eliangiringa Kaale.

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
