## [Decision Letter · Decision Letter 0]

14 Mar 2024

PGPH-D-24-00071

Strengthening Regulation for Medical Products in Tanzania: An assessment of Regulatory Capacity Development, 1978–2020

Dear Dr. Kaale,

Thank you for submitting your manuscript to PLOS Global Public Health. After careful consideration, we feel that it has merit but does not fully meet PLOS Global Public Health’s publication criteria as it currently stands. Therefore, we invite you to submit a revised version of the manuscript that addresses the points raised during the review process.

The reviews were, in general, favorable. However, the referees find that major revisions are necessary, and we agree with them. We would like you to address each of their concerns to the greatest extent as possible. Please, note that your manuscript will be subject to re-review before a decision is rendered.

In addition, please verify the data disclosure guidelines to ensure they have been correctly addressed. 

We look forward to receiving your revised manuscript.

Kind regards,

Elize Massard da Fonseca, Ph.D.

Academic Editor

Journal Requirements:

1. Please ensure that Funding Information and Financial Disclosure Statement are matched.

2. In the Funding Information you indicated that no funding was received. Please revise the Funding Information field to reflect funding received.

3. Please provide separate figure files in .tif or .eps format only and remove any figures embedded in your manuscript file. Please also ensure all files are under our size limit of 10MB.

4. We do not publish any copyright or trademark symbols that usually accompany proprietary names, eg  ©, ®, ™  (e.g. next to drug or reagent names). Please remove all instances of trademark/copyright symbols throughout the text, including ® on pages 13, 17 & 45.

Additional Editor Comments (if provided):

Reviewers' comments:

Reviewer's Responses to Questions

**Comments to the Author**

1. Does this manuscript meet PLOS Global Public Health’s publication criteria? Is the manuscript technically sound, and do the data support the conclusions? The manuscript must describe methodologically and ethically rigorous research with conclusions that are appropriately drawn based on the data presented.

Reviewer #1: Partly

Reviewer #2: Partly

Reviewer #3: Yes

2. Has the statistical analysis been performed appropriately and rigorously?

Reviewer #1: N/A

Reviewer #2: N/A

Reviewer #3: N/A

3. Have the authors made all data underlying the findings in their manuscript fully available (please refer to the Data Availability Statement at the start of the manuscript PDF file)?

Reviewer #1: No

Reviewer #2: No

Reviewer #3: Yes

4. Is the manuscript presented in an intelligible fashion and written in standard English?

Reviewer #1: No

Reviewer #2: No

Reviewer #3: Yes

5. Review Comments to the Author

Reviewer #1: Thank you for the opportunity to comment on this interesting paper. I’d like to commend the authors for providing a detailed overview of the changes in Tanzania’s regulatory landscape.

Major comments

1. The sections summarizing major challenges are not supported by any references and it is not clear how these challenges were identified. Were there systematic efforts to assess how the regulatory system works, e.g. regular assessments of regulatory resources and outcomes or surveys of stakeholders’ perceptions ? This seems to be suggested in the discussion section. IT’s important to mention this in the results and to show that there was a feedback loop informing further reforms. Also, some of the major challenges have been not addressed by subsequent reforms. This should be also noted among the results.

2. The current structure (historical context, major challenges, how challenges were addressed) works quite well but there is a lot of repetition which could be reduced by re-organising the material. E.g., lack of independence and autonomy included under ‘Medicines regulatory capacity’ but to a certain extent already dealt with in the previous section ‘Medicines regulatory framework’. Paragraphs in the section ‘ Medicines promotion and advertising’ overlap with those in the section ‘Market surveillance and control.’

3. EAC regional harmonization is mentioned but it is not clear how this initiative impacted regulatory capacities in Tanzania. The aim of the harmonization is to use regulatory resources more efficiently through workload sharing, e.g. joint inspections. Similarly, other initiatives such as WHO collaborative registration procedure aim to reduce duplication and bring information sharing which is important for regulatory strengthening.

please see the attached file for more detailed comments

Reviewer #2: Thank you for the opportunity to review the attached “Strengthening Regulation for Medical Products in Tanzania: An Assessment of Regulatory Capacity Development, 1978–2020”. I found this piece detailing the evolution of regulatory capacity in Tanzania very interesting and worthwhile, although there are significant issues that I recommend need to be addressed prior to being considered for publication.

Firstly, I encountered difficulty reconciling the objectives outlined in the abstract and introductory sections with the actual content and main findings of the article. For instance, the abstract stated that ”we analyse changes in the regulation of the Tanzania pharmaceutical sector over three main periods that map to points of significant change in the sector “; and then, “The study identified three key periods of transformation in Tanzania's pharmaceutical regulation: 1) The separation of medicines regulation from food safety (1978 to 2003), 2) The expansion of regulatory domains and the establishment of a semi-autonomous regulatory agency (2003–2011), and 3) The expanded role of the Pharmacy Council, including premise regulation (2011 to 2020)”. However, these three key periods have not been explicitly developed elsewhere. While the evolution of the legal framework implies the existence of these phases, a comprehensive elaboration of how changes to the regulatory framework, regulatory capacities, and core functions vary throughout each period is warranted in the results section. Similarly, the information presented in Tables 1, 3, and 4 is inconsistent with these three temporal stages highlighted. Furthermore, the authors stated, “This led us to ask how such capacity was developed and whether that history held implications for other countries wishing to strengthen their own medicines regulatory capacity.” However, the article neglects to delve into the mechanisms and rationale behind how the Tanzanian regulatory system's evolution and effectiveness could serve as a model for informing policy to strengthen regulatory frameworks in similar contexts.

Secondly, and the much more pressing issue, is the contribution of this study. I struggle to see how this research will significantly impact the literature on regulatory system strengthening beyond what has been previously documented. Notably, factors such as agency independence, accountability, and the influence of political, financial, and infrastructural elements have been extensively emphasized as pivotal for advancing toward an effective regulatory system (See, for example Shi, J., Chen, X., Hu, H., & Ung, C. O. L. (2023). Application of implementation science framework to develop and adopt regulatory science in different national regulatory authorities. Frontiers in Public Health, 11, 1172557; Regulatory System Strengthening in the Americas. Lessons Learned from the National Regulatory Authorities of Regional Reference. Washington, D.C.: Pan American Health Organization (2021); Chahal HS, Kashfipour F, Susko M, Feachem NS, Boyle C. Establishing a regulatory value chain model: An innovative approach to strengthening medicines regulatory systems in resource-constrained settings. Rev Panam Salud Publica. 2016;39(5):299–305).

Furthermore, the interpretation of results in the discussion section should encompass an analysis of how these findings augment or align with existing understandings of the issue. Given the article's focus on elucidating factors influencing the evolution of Tanzania's medicines regulatory authority, an examination from the perspective of institutional change theory could provide valuable insights (see: Coccia, Mario, An Introduction to the Theories of Institutional Change (2018). Journal of Economics Library, vol. 5, n. 4, pp. 337-344, 2018).

Finally, in response to “how such capacity was developed”, the identification of “3) Continuous capacity building and regulatory systems strengthening” in the discussion section, seems somewhat tautological to the research premise.

Thirdly, there are significant issues concerning the structure of the article, as well as the methodological design and analytical framework, which warrant review:

- The structure and content of each section require careful review. While the introductory section initially outlines the significance of effective regulation and the challenges encountered in regulatory strengthening in sub-Saharan Africa, it also incorporates information that, in my view, pertains more to the findings.

For instance, “… They embarked on a series of capacity building interventions that included human resource capacity building, organizational systems strengthening, decentralization of operations, and the promotion of financial sustainability” …” The recorded change in improvement in regulatory system in Tanzania was not a standalone effort. Various development partners have contributed both technical assistance (TA) and financial assistance…”), seem to belong more to the findings section. Moreover, towards the end of the introduction, the authors state ,” The paper is organized as follows: In the next sections, we present a contextual background of the pharmaceutical sector in Tanzania, then we present the methods used in the study…”. However, instead of a dedicated subsection to contextualize the pharmaceutical sector in Tanzania, the introduction is followed by the Methodology section.

- From a methodological standpoint, the study would benefit from a thorough description of the processes involved in conducting the documentary analysis of government documents and scoping literature. This should include detailed explanations of the data retrieval and collection methods, the timeline followed, and the procedures for data extraction and analysis. Additionally, the authors should aim to acknowledge more comprehensively the limitations inherent in both the study design and the findings. On the other hand, there is a need for a clearer elaboration on how the analytical framework was developed. This would involve providing insights into the rationale behind the framework's design, its components, and how it was applied to analyze the data collected. For instance, the authors utilized the WHO's 2019 Global Benchmarking Tool, notwithstanding that a newer version was issued in 2021. However, they did not expound upon this choice or justify the criteria for considering certain core functions or their components as categories or subcategories for the analysis. Furthermore, there is a lack of explanation regarding how the new analytical framework incorporates key indicators for each core function, such as leadership and crisis management, transparency, accountability, and communication. Efforts should be directed towards refining and validating the analytical framework to effectively map the evolution of core elements of the national regulatory system (e.g., legal bases, standards, guides and procedures, resources, etc.) and its essential regulatory functions.

Some other issues the authors should consider as they rework the manuscript:

1) Findings: This is a comprehensive review of the evolution of Tanzania’s NRA, and there was an emphasis on distinguishing the development of legal instruments, regulatory capacities, and functions over three periods (i.e., 1978 to 2003; 2003 to 2011; 2011-2020). The authors could construct a narrative review to report their findings over each period. For instance, it is suggested that three sub-sections corresponding to each period contain different changes in each of the dimensions analyzed.

2) Tables and figures: As a general recommendation for the authors, please consider spelling out abbreviations at first mention in tables or consider defining the abbreviations in the table note or figure caption, respectively, even though the abbreviations have been previously defined in the main text.

4.1 Table 1: In my opinion, table 1 is misplaced in the Introduction section as the reader has not been clearly presented with the regulatory functions, and it has not been explained at this point why the variations in regulatory oversight are reported across three different time periods. It seems better suited for the Findings sub-section. Also, the authors need to clarify better what the term “Regulatory Function” refers to in this table and how it differs from the regulatory functions presented in Table 2 and Table 5; and described in the Methodology section: “We defined regulatory functions to include the major responsibilities of a medicines regulatory authority as described by WHO 3 30 39 These include 1) National Regulatory Systems, 2) Medicines registration and marketing authorization, 3) vigilance, premises licensing, 4) regulatory inspection, 5) market surveillance and control, 6) clinical trials oversight, and 7) laboratory access and testing”. More suited terms for Table 1 might be “jurisdictional assignment” or “scope of regulatory oversight”.

4.2 Table 2: Heading “Analytic Categories” could be replaced by “Categories of analysis”.

Please include references in each category.

4.3 Table 3. It is unclear why this table only presents the results over two separate periods (i.e., 1978 to 2003; and 2003 to 2020) instead of three periods. This is confusing and makes it difficult for the reader to detect the key changes in the Tanzanian regulatory framework (e.g., The Pharmacy Act of 2002, the Tanzania Food, Drugs and Cosmetics Act of 2003, the Pharmacy Act of 2011). Additionally, please carefully review using “not existing” to describe the quality and risk management approach before the Tanzania Food, Drugs and Cosmetics Act of 2003 enactment, as it oversimplifies how (although incipient) quality assurance was regulated before the 2003 Act or subsequent regulations. Finally, please consider aligning the “Domains” described with those “Regulatory functions” outlined in Table 1. For instance, in Table 1 Retail and wholesale premises licensing falls under the Pharmacy Council from 2003 onwards, but this “domain” is not included in Table 3 under Pharmacy Council.

4.4 Table 4: Please see my comment above on using two instead of three periods to report the findings. New information such as the number of staff members and technology access not presented in the article body needs to include a reference or source of information. Also, please explain how “Low domestic production capacity,” -which seems to be a characteristic of the local pharmaceutical market, represented a challenge for the regulatory capacity between 2003 and 2020.

4.5. Table 6: Please add sources of information.

3) Overall comment about grammar and style: there are many grammatical mistakes, and reading can be awkward due to sentence structure and style. Additionally, important references are missing. For example, the fifth paragraph of the introduction lacks proper citations for the information regarding the characteristics of the pharmaceutical sector; and the definition of categories Regulatory Framework and Regulatory Capacity lack references in both text and Table 2.

Again, thank you for the opportunity to review this manuscript. I hope my suggestions are useful to the authors.

Reviewer #3: This article explores the development of pharmaceutical regulatory capacity in Tanzania. It offers a clear organizational scheme and analytic structure, and lots of systematic evidence across analytic categories of interest to document its findings.

Some larger and smaller suggestions would enhance the analytic value and utility of the paper, as well as its clarity. I would encourage the authors to pursue them.

-- Larger suggestions

The analytic framing and implications could use some context and development. Here area a few questions and a few related pieces of scholarship to engage, all of which would help in these areas, particularly in the introduction and discussion/conclusion:

Do all national pharmaceutical regulatory bodies go through a similar process over time? What’s unique about this one? If yes, why, if not, why not?

In what national regulatory body cases wouldn’t the components in the introduction and/or on page 22 obtain?

What makes the case of Tanzania unique in general, and/or in Africa?

When would we not expect the kinds of improvements over time that we see here?

What’s the politics behind all of this, and at different points in time?

On the general framework and history, it’d be worth engaging a range of literature, all of which are described below.

First is Dan Carpenter’s book, including the chapter(s) (9? 11?) on comparative cases:

Carpenter, D., 2010. Reputation and power: Organizational image and pharmaceutical regulation at the FDA. In Reputation and Power. Princeton University Press.

On further comparative components, this edited book by Greer/King/Massard da Fonseca/Peralta-Santos would be helpful:

Greer, Scott L., Elizabeth King, Elize Massard da Fonseca, and Andre Peralta-Santos. Coronavirus politics: The comparative politics and policy of COVID-19. University of Michigan Press, 2021.

Likewise, a recent special issue of JHPPL edited by Jarman/Massard da Fonseca/King would be helpful in various ways:

https://read.dukeupress.edu/jhppl/issue/49/1

Particularly so the article by the Editors, and by Nachlis and Thomson, particularly on the conclusions and recommendations issue:

Jarman, Holly, Elize Massard da Fonseca, and Elizabeth J. King. "The Political Economy of Vaccines during the COVID-19 Pandemic." Journal of Health Politics, Policy and Law 49, no. 1 (2024): 1-8.

Nachlis, Herschel, and Kyle Thomson. "Emergency Regulatory Procedures, Pharmaceutical Regulatory Politics, and the Political Economy of Vaccine Regulation in the COVID-19 Pandemic." Journal of Health Politics, Policy and Law 49, no. 1 (2024): 73-98.

On the points about weaknesses in pharmaceutical regulatory structures amidst growing strengths, see the Nachlis article here:

Nachlis, Herschel. "Pockets of weakness in strong institutions: Post-marketing regulation, psychopharmaceutical drugs, and medical autonomy, 1938–1982." Studies in American Political Development 32, no. 2 (2018): 257-291.

-- Smaller suggestions

- Table 1 is confusing – the columns are labeled as years but describe organizations.

- Table 6 – the numbers don’t add up to the total.

- Table 6 – was there really $8 billion in support? That seems high, but perhaps it is not.

- Figure 1 isn’t very clear, or at least the first 2/3 of it (the bottom 1/3 is more helpful).

- The writing becomes redundant at times in a range of places, and these portions could be cut down to reduce length and enhance clarity.

- The analysis can be quite general at times in a range of places, and these portions could likewise be cut down to reduce length and enhance clarity.

- Some of the descriptors are a bit confusing in some places (for example, “clamours” and “saddled”)

Again, overall, this is a very and very helpful article, with lots of good information, that could also benefit from all of the preceding suggestions.

6. PLOS authors have the option to publish the peer review history of their article (what does this mean?). If published, this will include your full peer review and any attached files.

**Do you want your identity to be public for this peer review?** For information about this choice, including consent withdrawal, please see our Privacy Policy.

Reviewer #1: No

Reviewer #2: No

Reviewer #3: No

---

## [Decision Letter · Decision Letter 1]

3 Jul 2024

PGPH-D-24-00071R1

Strengthening Regulation for Medical Products in Tanzania: An assessment of Regulatory Capacity Development, 1978–2020

Dear Dr. Kaale,

Thank you for submitting your manuscript to PLOS Global Public Health. After careful consideration, we feel that it has merit but does not fully meet PLOS Global Public Health’s publication criteria as it currently stands. Therefore, we invite you to submit a revised version of the manuscript that addresses the points raised during the review process.

Strengthening regulation for medical products in low- and middle-income countries is undoubtedly a timely and relevant topic. I appreciated the empirical information presented. However, the manuscript still requires substantial revision. Given the topic's relevance, I am willing to provide you with one final opportunity for revisions.

In the abstract and conclusion, you mention the importance of international partners such as the Global Fund, UNDP, and MSH. However, the analysis fails to explain how these organizations contributed to the process of strengthening regulation in Tanzania.As mentioned by R1 and R2, the response to the "so what" question has not been adequately addressed yet. What does this manuscript add to our knowledge about regulatory system strengthening?R1 still has some crucial comments that need to be addressed: better presentation of the evidence, formatting, and editing of the manuscript. The fact that there is a paragraph duplicated within the introduction suggests that the manuscript was not carefully revised.

Please note that this is your last chance to make the necessary revisions. Failure to adequately address these points will result in the rejection of your manuscript.

We look forward to receiving your revised manuscript.

Kind regards,

Elize Massard da Fonseca, Ph.D.

Academic Editor

Journal Requirements:

1. We do not publish any copyright or trademark symbols that usually accompany proprietary names, eg (R), (C), or TM  (e.g. next to drug or reagent names). Please remove all instances of trademark/copyright symbols throughout the text, including ® on pages 17, 21 and 49.

Additional Editor Comments (if provided):

Reviewers' comments:

Reviewer's Responses to Questions

**Comments to the Author**

1. If the authors have adequately addressed your comments raised in a previous round of review and you feel that this manuscript is now acceptable for publication, you may indicate that here to bypass the “Comments to the Author” section, enter your conflict of interest statement in the “Confidential to Editor” section, and submit your "Accept" recommendation.

Reviewer #1: (No Response)

Reviewer #2: All comments have been addressed

2. Does this manuscript meet PLOS Global Public Health’s publication criteria? Is the manuscript technically sound, and do the data support the conclusions? The manuscript must describe methodologically and ethically rigorous research with conclusions that are appropriately drawn based on the data presented.

Reviewer #1: Partly

Reviewer #2: Yes

3. Has the statistical analysis been performed appropriately and rigorously?

Reviewer #1: N/A

Reviewer #2: N/A

4. Have the authors made all data underlying the findings in their manuscript fully available (please refer to the Data Availability Statement at the start of the manuscript PDF file)?

Reviewer #1: No

Reviewer #2: Yes

5. Is the manuscript presented in an intelligible fashion and written in standard English?

Reviewer #1: No

Reviewer #2: Yes

6. Review Comments to the Author

Reviewer #1: PLOS Global Public Health

R1: Strengthening Regulation for Medical Products in Tanzania: An assessment of Regulatory Capacity Development, 1978–2020

Thank you for the revision which addressed the majority of reviewers’ comments (some answered in the response to reviewers document but do not seem to be sufficiently addressed in the actual manuscript). The revision still leaves some sections in disarray and it is a challenge to read the manuscript in one go a remain engaged. The authors are trying to summarise a lot of material in this manuscript and I would like to ask them to carefully read and edit the manuscript with their audience in mind.

Please see some specific examples below (but the whole manuscript needs a careful editing and checking for consistency).

Abstract

Background – ‘for well-functioning of the regulatory system’ ... why not “a well-functioning regulatory system” ? (here and later in the manuscript)

Results, 1st sentence – there’s redundant ‘of’ before ‘the regulatory capacity

Results, 2nd sentence – ‘under mirror’ ? do you mean ‘under the microscope’ ; the following ‘correspond to’ is redundant; ‘and’ before the point 2) should be moved to before the point 3)

Conclusions – the first sentence need a full stop at the end and a space to be added before the 2nd sentence.

Delete ‘, and others’ at the end of the 2nd para.

The 3rd para. Does not follow from the results presented (here and in the conclusion).

Introduction

Para.5 – ‘Effective medicine regulation’ ? was it effective or active?

Para. 6 – ‘as well as organizational structures’ ? under a different organizational structures; this part need a better formulation

Para. 8 – ‘However’ is not the right word to use here. This paragraph refers to the WHO guidance and recommendations in this area without mentioning the WHO. The next para then starts with ‘using this framework’ but it’s not clear what framework is being referred to.

Para. After “[insert table 1] – delete “In” and start with “A recent ...” In this sentence FDA is mentioned but the abbreviation is not explained. In the 2nd sentence it is not clear what “the latter” refers to. TO corona politics?

The next para. Repeats the sentence on Carpenter’s chapter

Penultimate paragraph – the outlined organization of the paper does not correspond to what follows

Methods, objectives - how do you assess the effectiveness of the regulatory system?

Analytical framework, 1st sentence – ‘our study objective’ should be plural ‘objectives’ ?

- 2nd para. It looks like that instead of regrouping you considered the above mentioned dimensions under regulatory functions and complemented with two other categories (re. Framework, reg. Capacity)

Data sources and collection – ‘conducted key informant interviews’ - more details needed here on how many, when, focus of the interviews (structured, open end questions, etc), how analysed; ethical approval required ?

The next sentence “The of data included...” needs correction

p.8 Major challenges – “in both periods analysed” – the abstract refers to three periods

and so on...

The discussion primarily provides a summary findings but does not relate to the broader literature. Here, the comment by Reviewer 2 is relevant – what does this manuscript add to what we already know about regulatory system strengthening?

Reviewer #2: The authors addressed the comments previously raised. Please still check the consistency of references. I detected this paragraph duplicated within the same section (introduction) and with different reference numbers: "In a recent book chapter by Carpenter explains how the FDA's reputation and power have inter played in shaping regulation36 and recent corona politics 37 39-41"

7. PLOS authors have the option to publish the peer review history of their article (what does this mean?). If published, this will include your full peer review and any attached files.

**Do you want your identity to be public for this peer review?** For information about this choice, including consent withdrawal, please see our Privacy Policy.

Reviewer #1: No

Reviewer #2: No

---

## [Editor Report · Decision Letter 2]

25 Sep 2024

Strengthening Regulation for Medical Products in Tanzania: An assessment of Regulatory Capacity Development, 1978–2020

PGPH-D-24-00071R2

Dear Professor Kaale,

We are pleased to inform you that your manuscript 'Strengthening Regulation for Medical Products in Tanzania: An assessment of Regulatory Capacity Development, 1978–2020' has been provisionally accepted for publication in PLOS Global Public Health.

Best regards,

Elize Massard da Fonseca, Ph.D.

Academic Editor